# Effect of weekend admission on process of care and clinical outcomes for the management of acute coronary syndromes: a retrospective analysis of three UK centres

Glen P Martin,[1] Tim Kinnaird,[2,3] Matthew Sperrin,[1] Richard Anderson,[3] Amr Gamal,[4] Avais Jabbar,[4] Chun Shing Kwok,[2,5] Diane Barker,[5] Grant Heatlie,[5] Azfar G Zaman,[4] Mamas A Mamas[1,2,5]

For numbered affiliations see end of article.

**Correspondence to**
Dr Mamas A Mamas;
mamasmamas1@yahoo.co.uk

## ABSTRACT

**Objectives** The effect of weekend versus weekday admission following acute coronary syndrome (ACS) on process of care and mortality remains controversial. This study aimed to investigate the 'weekend-effect' on outcomes using a multicentre dataset of patients with ST elevation myocardial infarction (STEMI) and non-ST elevation myocardial infarction/unstable angina (NSTEMI/UA).

**Design** This retrospective observational study used propensity score (PS) stratification to adjust estimates of weekend effect for observed confounding. Logistic regression was used to estimate odds ratios (ORs) for binary outcomes and time-to-event endpoints were modelled using Cox proportional hazards to estimate hazard ratios (HRs).

**Setting** Three tertiary cardiac centres in England and Wales that contribute to the Myocardial Ischaemia National Audit Project.

**Participants** Between January 2010 and March 2016, 17 705 admissions met the study inclusion criteria, 4327 of which were at a weekend.

**Primary and secondary outcomes** Associations were studied between weekend admissions and the following primary outcome measures: in-hospital mortality, 30-day mortality and long-term survival; secondary outcomes included several processes of care indicators, such as time to coronary angiography.

**Results** After PS stratification adjustment, mortality outcomes were similar between weekend and weekday admission across patients with STEMI and NSTEMI/UA. Weekend admissions were less likely to be discharged within 1 day (HR 0.72, 95% CI 0.66 to 0.78), but after 4 days the length of stay was similar (HR 0.97, 95% CI 0.90 to 1.04). Fewer patients with NSTEMI/UA received angiography between 0 and 24 hours at a weekend (HR 0.71, 95% CI 0.65 to 0.77). Weekend patients with STEMI were less likely to undergo an angiogram within 1 hour, but there was no significant difference after this time point.

**Conclusion** Patients with ACS had similar mortality and processes of care when admitted on a weekend compared with a weekday. There was evidence of a delay

## Strengths and limitations of this study

► An analysis of a weekend effect after acute coronary syndromes in contemporary practice.
► This study analysed data derived from a robust national database (Myocardial Ischaemia National Audit Project), of three large tertiary interventional centres.
► Included both ST elevation myocardial infarction cases and non-ST elevation myocardial infarction/unstable angina cases.
► Any analysis of a weekend effect will be confounded by patient demographics, healthcare systems and the hospitals involved, thus limiting the generalisability of the results.
► Due to the observational nature of the study, unmeasured confounders potentially influence the conclusions.

to angiography for patients with NSTEMI/UA admitted at the weekend.

## INTRODUCTION

A timely revascularisation following presentation with an acute coronary syndrome (ACS) can mitigate loss of cardiac function and reduce morbidity and mortality.[1 2] Invasive procedures such as primary percutaneous coronary intervention (PCI) are central to the care of patients presenting with ST elevation myocardial infarction (STEMI)[3] and guidelines also recommend an early invasive strategy for non-ST elevation myocardial infarction (NSTEMI) cases.[4 5]

Nevertheless, heterogeneity in hospital resource allocation, service provision and staffing levels across weekday and weekends can affect the delivery of optimal care, leading to studies of the so-called 'weekend effect'.[6–8]

Higher rates of mortality have been indicated in patients admitted to hospital for an acute myocardial infarction (AMI) at a weekend, an evening or holiday period.[7 9 10] However, previous results in patients with AMI are inconsistent.[11–13] An analysis of a multicentre registry highlighted that, despite weekend admission being associated with a delay in invasive therapy, this did not impact rates of adverse events.[13] Similarly, the 'Get With the Guidelines-Coronary Artery Disease' database showed that despite longer door-to-balloon time, mortality rates were similar between 'out-of-hour' and 'in-hour' hospital admission.[11]

Many previous studies have focused on outcome differences across weekend/weekday admissions in patients with STEMI, with limited data for patients with NSTEMI or unstable angina (NSTEMI/UA). Service provision/delivery, healthcare systems, deprivation and population demographics vary both temporally and by geography, thereby confounding the analysis of a weekend effect; even the definition of weekend or 'out-of-hours' admission varies between populations. Such factors might explain, in part, the inconsistencies across previous studies. Therefore, the aim of this study was to examine the effect of weekend versus weekday admission on outcomes following presentation with STEMI or NSTEMI/UA, in a contemporary, multicentre dataset of the UK.

## METHODS
### MINAP dataset
The Myocardial Ischaemia National Audit Project (MINAP) collects prospective data on the management of heart attacks in the UK.[14] Each centre is responsible for data entry into MINAP, based on agreed definitions and options for each variable. This study had access to data from three contributing centres, namely, the Freeman Hospital (Newcastle-upon-Tyne), the Royal Stoke Hospital (Stoke-on-Trent) and the University Hospital of Wales (Cardiff). The dataset includes variables detailing patient characteristics, emergency response/admission dates, processes of care and outcomes within hospital discharge. Long-term mortality tracking was available from the Office for National Statistics for English patients and the Welsh Demographic Service for Welsh patients.

### Study design
This retrospective analysis included all patients who had a discharge diagnosis of an ACS and were subclassified into STEMI, or NSTEMI/UA. After such inclusion criteria, the STEMI subgroup was defined as those patients with an admission and/or discharge diagnosis of STEMI; all other patients meeting the discharge diagnosis inclusion criteria were included in the NSTEMI/UA subgroup.

Using hospital admission dates, patients were classified into weekend or weekday admission (reference group). In line with previous studies, weekends were defined as any admission between 00:00 Saturday and 23:59 Sunday, with weekday defined as any admissions outside this time window.[7–9] The primary outcome measures were in-hospital mortality, 30-day mortality and long-term survival. Secondary outcomes included the following process of care indicators: admission to a cardiology ward, length of stay (defined as whole number of days between admission date and discharge date), performance of coronary angiography/echocardiography, time to coronary angiography (defined as number of hours between admission time and time of angiogram) and discharged on beta-blocker, ACE-inhibitors (ACE-I) or statins. Note that any patient admitted and discharged on the same day had length of stay defined as zero. All the outcomes were analysed as a whole cohort (with adjustment for STEMI indication) and separately in the STEMI and NSTEMI/UA subgroups.

### Statistical analysis
For descriptive analysis, continuous variables were presented as means with SD and compared using the t-test. Categorical variables were presented as frequencies of occurrence and compared using the $X^2$ test. Patients with missing endpoint variables were excluded from the analysis of that endpoint. Other patient characteristic variables that were missing were assigned to the gender-matched median value for continuous variables or the reference category for categorical variables.

To control for potential differences in baseline covariates between weekend and weekday admission, propensity scores (PS) for being admitted at a weekend were calculated for each patient.[15 16] Although day of hospital admission is not a modifiable intervention, one could regard this in the context of differences in resource allocation between the weekend and weekday. Variables included in the logistic regression model to calculate the PS included all those given in table 1 and an admitting centre indicator. Patients were stratified into five strata using their PS, with the cut-off values determined by the quintiles of the PS distribution for all patients. Such PS stratification has been shown to remove 90% of the bias due to the covariates within the PS model.[15 16]

Within each PS strata, binary outcomes (mortality, admission to cardiology ward, performance of coronary angiography/echocardiography and prescription of medications) were compared directly between admission day groups (using logistic regression), with the odds ratios (ORs) pooled by the Mantel-Haenszel method.[17] For time-to-event outcomes (length of stay, time-to-angiography and long-term survival), the unadjusted survival curves were obtained using the Kaplan-Meier estimate and adjusted hazard ratios (HRs) were estimated using a Cox proportional hazards model. The Cox proportional hazards model included weekend indication as the covariate and was stratified by PS strata. The proportional hazards assumption of the weekend effect variable was checked by examining the Schoenfeld residuals against time; weekend-by-time interactions were included where necessary. We did not model length of stay beyond 50 days, or time-to-angiography beyond 7 days for NSTEMI/

**Table 1** Baseline summary

| Variable | Whole cohort (n=17 705) | Weekday admission (n=13378) | Weekend admission (n=4327) | p-Value | Missing (% of whole cohort) |
|---|---|---|---|---|---|
| Age at admission, mean (SD) | 66.0 (13.0) | 66.2 (13.0) | 65.5 (13.1) | 0.001 | 7 (0.04) |
| Male, n (%) | 12 530 (70.8) | 9410 (70.3) | 3120 (72.1) | 0.028 | 0 (0.00) |
| Caucasian, n (%) | 12283 (69.4) | 9360 (70.0) | 2923 (67.6) | 0.003 | 4988 (28.2) |
| **Admission diagnosis** | | | | | |
| Acute coronary syndrome, n (%) | 15 427 (87.1) | 11 689 (87.4) | 3738 (86.4) | 0.097 | 0 (0.00) |
| Chest pain unknown cause, n (%) | 1759 (9.94) | 1316 (9.84) | 443 (10.2) | 0.461 | |
| Other, n (%) | 519 (2.93) | 373 (2.79) | 146 (3.37) | 0.053 | |
| Previous AMI, n (%) | 3768 (21.3) | 2907 (21.7) | 861 (19.9) | 0.011 | 109 (0.62) |
| Previous angina, n (%) | 3968 (22.4) | 3073 (23.0) | 895 (20.7) | 0.002 | 282 (1.59) |
| Hypertension, n (%) | 9392 (53.0) | 7153 (53.5) | 2239 (51.7) | 0.050 | 255 (1.44) |
| Hypercholesterolaemia, n (%) | 7166 (40.5) | 5458 (40.8) | 1708 (39.5) | 0.127 | 574 (3.24) |
| PVD, n (%) | 809 (4.57) | 626 (4.68) | 183 (4.23) | 0.234 | 85 (0.48) |
| Cerebrovascular disease, n (%) | 1270 (7.17) | 998 (7.46) | 272 (6.29) | 0.010 | 83 (0.47) |
| Asthma or COPD, n (%) | 2717 (15.3) | 2095 (15.7) | 622 (14.4) | 0.044 | 114 (0.64) |
| Chronic renal failure, n (%) | 691 (3.90) | 527 (3.94) | 164 (3.79) | 0.693 | 143 (0.81) |
| Heart failure, n (%) | 508 (2.87) | 379 (2.83) | 129 (2.98) | 0.649 | 159 (0.90) |
| Enzymes elevated, n (%) | 16378 (92.5) | 12 304 (92.0) | 4074 (94.2) | <0.001 | 634 (3.58) |
| Previous/current smoker, n (%) | 11 892 (67.2) | 9021 (67.4) | 2871 (66.4) | 0.195 | 550 (3.11) |
| Cholesterol, mean (SD) | 4.77 (1.35) | 4.76 (1.35) | 4.82 (1.38) | 0.015 | 4590 (25.9) |
| **Diabetes** | | | | | |
| Dietary control, n (%) | 609 (3.44) | 464 (3.47) | 145 (3.35) | 0.749 | 126 (0.71) |
| Oral medicine, n (%) | 1870 (10.6) | 1394 (10.4) | 476 (11.0) | 0.293 | |
| Insulin, n (%) | 721 (4.07) | 571 (4.27) | 150 (3.47) | 0.023 | |
| Insulin and medication, n (%) | 297 (1.68) | 224 (1.67) | 73 (1.69) | 0.999 | |
| Previous PCI, n (%) | 2124 (12.0) | 1651 (12.3) | 473 (10.9) | 0.014 | 17 (0.10) |
| Previous CABG, n (%) | 959 (5.42) | 732 (5.47) | 227 (5.25) | 0.595 | 4 (0.02) |
| Systolic BP, mean (SD) | 135.3 (27.7) | 135.1 (27.5) | 135.8 (28.3) | 0.219 | 2574 (14.5) |
| Heart Rate, mean (SD) | 76.8 (19.3) | 76.5 (19.1) | 77.5 (19.8) | 0.006 | 2257 (12.7) |
| **Admitting consultant** | | | | | |
| Cardiologist, n (%) | 17 093 (96.5) | 12 929 (96.6) | 4164 (96.2) | 0.216 | 52 (0.29) |
| Other general physician, n (%) | 445 (2.51) | 325 (2.43) | 120 (2.77) | 0.230 | |
| Other, n (%) | 115 (0.65) | 88 (0.66) | 27 (0.62) | 0.895 | |
| Beta blocker, n (%) | 4670 (26.4) | 3555 (26.6) | 1115 (25.8) | 0.306 | 617 (3.48) |
| Statin, n (%) | 6997 (39.5) | 5360 (40.1) | 1637 (37.8) | 0.009 | 630 (3.56) |
| Glucose, mean (SD) | 8.60 (4.60) | 8.55 (4.61) | 8.76 (4.58) | 0.015 | 3044 (17.2) |
| Height, mean (SD) | 170.1 (9.76) | 170.0 (9.81) | 170.5 (9.57) | 0.011 | 3514 (19.8) |

Continued

UA and beyond 5 hours for STEMI. All estimates are reported with corresponding 95% CIs.

R V.3.3.1[18] was used for all statistical analyses. Graphical plots were made using the 'ggplot2' package[19] and the 'survival' package was used for time-to-event analyses.[20][21]

## RESULTS

There were 18 166 admissions across the three centres between 1 January 2010 and 30 March 2016. This comprised 3140 admissions in University Hospital of Wales, 7023 admissions in Royal Stoke Hospital and 8003 admissions in Freeman Hospital. Together, 461 admissions did not meet the discharge diagnosis inclusion criteria. Thus, the analysis sample size was 17 705 with n=9322 admissions for a STEMI and n=7887 admissions for NSTEMI/UA; the type of ACS was unknown in the remaining 496 records, which were removed from the STEMI and NSTEMI/UA subgroup analyses.

Table 1 gives the patient baseline characteristics by weekend or weekday admission. Patients admitted on a weekday were older (p=0.001), were more likely female (p=0.028) and had higher rates of previous AMI (p=0.011), previous angina (p=0.002), cerebrovascular disease (p=0.010), asthma/chronic obstructive pulmonary disease (p=0.044) and previous PCI (p=0.014). Conversely, weekend patients were more likely to have elevated enzymes (p<0.001) or present with ST elevation (p<0.001), although Killip class did not significantly differ between admission groups. Importantly, rates of admission to a cardiologist were not different across weekend or weekday cohorts, with both being over 96% (p=0.216).

### Mortality

Figure 1 shows hospital mortality rates by day of the week. There was no apparent trend in either overall hospital mortality or cardiac/non-cardiac related deaths. In the whole cohort, the observed hospital mortality rate was 3.96% and 3.41% for weekday and weekend admission groups, respectively (OR 0.86, 95% CI 0.71 to 1.03) (table 2). After PS stratification, the pooled OR was 0.82 (95% CI 0.58 to 1.09). Similar findings were found across STEMI and NSTEMI/UA subgroups (table 3).

Overall, 1-year, 3-year and 5-year survival rates were 89.7%, 82.3% and 74.9%, respectively. As a whole cohort, there was no significant difference in the Kaplan-Meier survival curves between admission groups (log-rank test, p=0.569) (figure 2). The PS-adjusted HR from a Cox proportional hazards model for the whole cohort was 0.99 (95% CI 0.90 to 1.09) for weekend over weekday admission. In STEMI cases, the Kaplan-Meier curves were significantly different (log-rank test, p=0.032), but this was not observed for NSTEMI/UA cases (log-rank test, p=0.103). After PS adjustment, long-term survival was not significantly different between weekday or weekend admissions, with a HR for STEMI cases of 0.94 (95% CI 0.82 to 1.07) and for NSTEMI/UA cases of 1.02 (95% CI 0.89 to 1.18).

**Table 1** Continued

| Variable | Whole cohort (n=17 705) | Weekday admission (n=13 378) | Weekend admission (n=4327) | p-Value | Missing (% of whole cohort) |
|---|---|---|---|---|---|
| Weight, mean (SD) | 81.9 (18.7) | 81.8 (18.8) | 82.0 (18.3) | 0.624 | 2571 (14.5) |
| Family history of CHD, n (%) | 8287 (46.8) | 6273 (46.9) | 2014 (46.5) | 0.705 | 1939 (11.0) |
| Creatinine, mean (SD) | 95.7 (64.6) | 95.5 (63.3) | 96.3 (68.4) | 0.502 | 266 (1.50) |
| Haemoglobin, mean (SD) | 136.6 (19.1) | 136.3 (19.2) | 137.5 (18.8) | <0.001 | 195 (1.10) |
| Killip class | | | | | 6919 (39.1) |
| No evidence of heart failure, n (%) | 9103 (51.4) | 6918 (51.7) | 2185 (50.5) | 0.170 | |
| Basal crepitations, n (%) | 811 (4.58) | 590 (4.41) | 221 (5.11) | 0.062 | |
| Pulmonary oedema, n (%) | 600 (3.39) | 453 (3.39) | 147 (3.40) | 0.999 | |
| Cardiogenic shock, n (%) | 266 (1.50) | 195 (1.46) | 71 (1.64) | 0.430 | |
| Cardiac arrest, n (%) | 1493 (8.43) | 1106 (8.27) | 387 (8.94) | 0.174 | 0 (0.00) |
| Thienopyridene, n (%) | 15632 (88.3) | 11774 (88.0) | 3858 (89.2) | 0.043 | 109 (0.62) |
| STEMI, n (%) | 9322 (52.7) | 6781 (50.7) | 2541 (58.7) | <0.001 | 496 (2.80) |

AMI, acute myocardial infarction; BP, blood pressure; CABG, coronary artery bypass graft; CHD, coronary heart disease; COPD, chronic obstructive pulmonary disease; LVEF, left ventricular ejection fraction; MI, myocardial infarction; PCI, percutaneous coronary intervention; PVD, peripheral vascular disease; STEMI, ST segment elevation myocardial infarction.

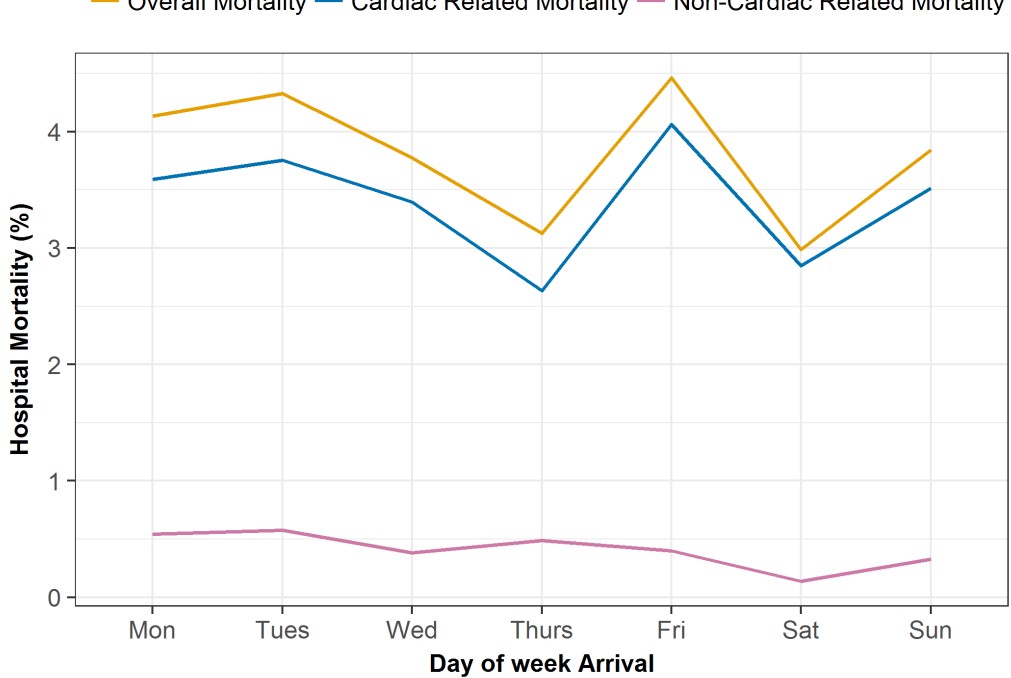

**Figure 1** In-hospital mortality rates by the day of arrival decomposed into cardiac and non-cardiac related mortality.

## Admission process

In the weekend group, 4045 of 4318 (93.7%) patients were admitted to either a cardiac ward or a coronary care unit, compared with 12 548 of 13 345 (94.0%) in the weekday group (table 2). Rates of admission to a cardiac ward or coronary care unit were not significantly different between the weekend and weekday, with an adjusted OR of 0.97 (95% CI 0.76 to 1.23). In STEMI admissions, the adjusted OR was 1.08 (95% CI 0.67 to 1.52) and in NSTEMI/UA admissions was 0.93 (95% CI 0.71 to 1.20) (table 3).

There was a significant difference in the Kaplan-Meier curves for length of stay between weekday and weekend admission for patients with STEMI (log-rank p=0.001) and for NSTEMI/UA (log-rank p<0.001) (see figure 1 in the online supplementary file 1). Weekend admission was associated with significantly lower hazards of being discharged within 1 day (HR 0.72, 95% CI 0.66 to 0.78), but significantly higher hazards of being discharged between 1 and 4 days (HR 1.13, 95% CI 1.08 to 1.18) (table 4). Similar findings were found when examining length of stay across STEMI and NSTEMI/UA subgroups (see table 1 in the online supplementary file 1).

## Coronary angiography

There was no significant difference in the proportion of patients receiving a coronary angiography between weekday and weekend admission (OR 0.91, 95% CI 0.71 to 1.13) (table 2). In STEMI cases, 96.5% and 96.6% of patients underwent an angiography on a weekday and weekend, respectively (adjusted OR 0.95, 95% CI 0.62 to 1.36). Similarly, patients with NSTEMI/UA patients did

not have significantly different odds of undergoing a coronary angiography by weekend admission, after PS stratification adjustment (table 3).

Within the group of patients who underwent a coronary angiography, we assessed the time-to-angiography where time of the angiogram was also available (n=7284 STEMI and n=5705 NSTEMI/UA). There was a significant difference in the cumulative event rates for time-to-angiography for NSTEMI/UA (log-rank p<0.001) and for STEMI cases (log-rank p=0.010) (figure 3). In NSTEMI/ UA cases, the number of patients receiving angiography within 24 hours differed significantly by weekend admission groups, with similar findings at 72 hours (see table 2 in the online supplementary file 1). For instance, 88.9% of weekend NSTEMI/UA admissions received an angiography within 72 hours, compared with 91.1% of weekday NSTEMI/ UA admissions (p=0.028). After PS adjustment, the time-dependent Cox proportional hazards model indicated that weekend patients with NSTEMI/UA were less likely to undergo a coronary angiography within 2 hours (HR 0.54, 95% CI 0.42 to 0.69) and between 2 and 24 hours (HR 0.71, 95% CI 0.65 to 0.77), with the observed mean time-to-angiography being approximately 3 hours longer within the weekend cohort by 24 hours (table 5). Baseline characteristics of patients with NSTEMI/UA between those undergoing angiography within 24 hours and those after 24 hours are given in table 3 in the online supplementary file 1. Patients with NSTEMI/UA who underwent angiography within 24 hours were more likely to have an ACS admission diagnosis (p<0.001), previous angina (p=0.024), hypercholesterolaemia (p<0.001) or

**Table 2** Event numbers, unadjusted OR and pooled OR across propensity score (PS) strata.

| Outcome | Weekday admission (n=13378) | Weekend admission (n=4327) | Unadjusted OR (95% CI) | PS adjusted OR (95% CI) |
|---|---|---|---|---|
| Hospital mortality | 529/13354 (3.96%) | 147/4312 (3.41%) | 0.86 (0.71 to 1.03) | 0.82 (0.58 to 1.09) |
| 30-day mortality | 553/10874 (5.09%) | 182/3545 (5.13%) | 1.01 (0.85 to 1.20) | 0.96 (0.72 to 1.28) |
| Admission to cardiology ward | 12548/13345 (94.0%) | 4045/4318 (93.7) | 0.94 (0.82 to 1.09) | 0.97 (0.76 to 1.23) |
| Coronary angiography | 12401/13265 (93.5%) | 3987/4288 (93.0%) | 0.92 (0.81 to 1.06) | 0.91 (0.71 to 1.13) |
| Echocardiography | 5772/11262 (51.3%) | 2014/3666 (54.9%) | **1.16 (1.08 to 1.25)** | **1.16 (1.01 to 1.31)** |
| Discharged on beta-blocker | 10320/12181 (84.7%) | 3404/3943 (86.3%) | **1.14 (1.03 to 1.26)** | 1.06 (0.88 to 1.26) |
| Discharged on ACE-inhibitors | 10801/12109 (89.2%) | 3552/3930 (90.4%) | **1.14 (1.01 to 1.29)** | 1.09 (0.88 to 1.34) |
| Discharged on statins | 11590/12131 (95.5%) | 3764/3942 (95.5%) | 0.99 (0.83 to 1.18) | 0.96 (0.68 to 1.24) |

Bold items indicate significant effects at the 5% level.

**Table 3** Event numbers and pooled OR across propensity score (PS) strata by ST-elevation myocardial infarction (STEMI) and non-ST-elevation myocardial infarction/unstable angina (NSTEMI/UA) subgroups

| Outcome | STEMI (n=9322) | | | NSTEMI/UA (n=7887) | | |
|---|---|---|---|---|---|---|
| | Weekday admission (n=6781) | Weekend admission (n=2541) | PS-adjusted OR (95% CI) | Weekday admission (n=6190) | Weekend admission (n=1697) | PS-adjusted OR (95% CI) |
| Hospital mortality | 379/6774 (5.59%) | 110/2533 (4.34%) | 0.83 (0.58 to 1.15) | 137/6175 (2.22%) | 36/1691 (2.13%) | 0.82 (0.45 to 1.38) |
| 30-day mortality | 379/5633 (6.73%) | 132/2123 (6.22%) | 0.99 (0.69 to 1.35) | 157/4913 (3.20%) | 45/1352 (3.33%) | 0.88 (0.53 to 1.41) |
| Admission to cardiology ward | 6486/6763 (95.9%) | 2442/2533 (96.4%) | 1.08 (0.67 to 1.52) | 5673/6176 (91.9%) | 1518/1696 (89.5%) | 0.93 (0.71 to 1.20) |
| Coronary angiography | 6512/6749 (96.5%) | 2445/2530 (96.6%) | 0.95 (0.62 to 1.36) | 5532/6113 (90.5%) | 1465/1669 (87.8%) | 0.92 (0.70 to 1.15) |
| Echocardiography | 3241/6085 (53.3%) | 1267/2210 (57.3%) | 1.17 (0.98 to 1.38) | 2377/4796 (49.6%) | 714/1374 (52.0%) | 1.16 (0.93 to 1.42) |
| Discharged on beta-blocker | 5226/5976 (87.4%) | 2018/2261 (89.3%) | 1.08 (0.83 to 1.39) | 4838/5846 (82.8%) | 1328/1600 (83.0%) | 1.03 (0.80 to 1.32) |
| Discharged on ACE-inhibitor | 5529/5940 (93.1%) | 2109/2254 (93.6%) | 1.04 (0.72 to 1.41) | 4998/5815 (86.0%) | 1374/1594 (86.2%) | 1.13 (0.83 to 1.41) |
| Discharged on statins | 5772/5951 (97.0%) | 2187/2260 (96.8%) | 0.89 (0.50 to 1.32) | 5481/5824 (94.1%) | 1498/1600 (93.6%) | 1.01 (0.68 to 1.41) |

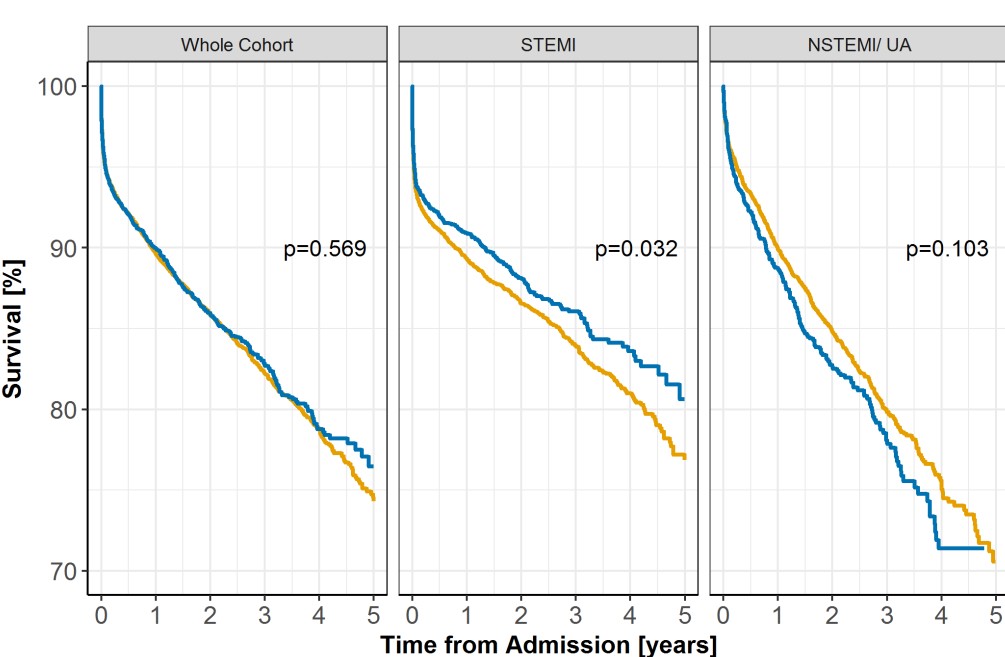

**Figure 2** Long-term survival following weekend versus weekday admission as a whole cohort and across ST-elevation myocardial infarction (STEMI) and non-ST-elevation myocardial infarction/unstable angina (NSTEMI/UA) subgroups.

a family history of coronary heart disease (p<0.001) and were significantly more likely to be admitted to a cardiologist (p<0.001) compared with those over 24 hours.

In patients with STEMI, after PS adjustment the Cox proportional hazards model indicated that there was a slight delay for weekend admission within 1 hour (HR 0.89, 95% CI 0.84 to 0.94) (table 5). Specifically, between 0 and 1 hour, the observed mean time-to-angiography was 0.55 hour and 0.57 hour for weekday and weekend admission, respectively. After 1 hour, there was no significant difference in time-to-angiography between weekend or weekday STEMI admissions after PS adjustment (table 5). Similarly, we examined door-to-balloon time in those patients with STEMI who underwent primary PCI as the initial reperfusion treatment and where time of reperfusion was also available. Here, door-to-balloon time was longer than 90 min (as recommended by the American College of Cardiology) in 6.95% of weekday admissions, compared with 9.12% of weekend admissions (p=0.004). Likewise, the European Society of Cardiology guideline of door-to-balloon time within 120 min was exceeded

significantly more in weekend admissions (4.13%) compared with weekday admissions (2.96%) (p=0.02).

### Echocardiography
At the weekend, 54.9% of patients underwent an echocardiogram, compared with 51.3% of weekday admissions, with both the unadjusted and PS-adjusted ORs being significantly different from one (table 2). Specifically, the pooled OR from the PS stratification was 1.16 (95% CI 1.01 to 1.31), indicating that patients admitted over the weekend had 16% greater odds of undergoing an echocardiography compared with those admitted during the week.

### Prescribed medication on discharge
The unadjusted analysis of both prescription of beta-blocker on discharge and prescription of ACE-I on discharge indicated significantly increased odds for those admitted at the weekend (table 2). However, these findings were not significant after adjusting for PS. The crude proportions of statins on discharge were similar across

**Table 4** Number of patients discharged within each time-window and propensity score (PS)-adjusted HRs from the time-dependent Cox proportional hazards models for length of stay, as a whole cohort.

| Length of stay | Discharged, n (%) weekday cohort | Discharged, n (%) weekend cohort | PS-adjusted HR (95% CI) |
|---|---|---|---|
| 0–1 day | 2888 (21.7%) | 636 (14.7%) | **0.72 (0.66 to 0.78)** |
| 1–4 days | 6731 (64.5%) | 2595 (70.5%) | **1.13 (1.08 to 1.18)** |
| 4–50 days | 3639 (98.2%) | 1063 (98.1%) | 0.97 (0.90 to 1.04) |

Bold items indicate significant effects at the 5% level.

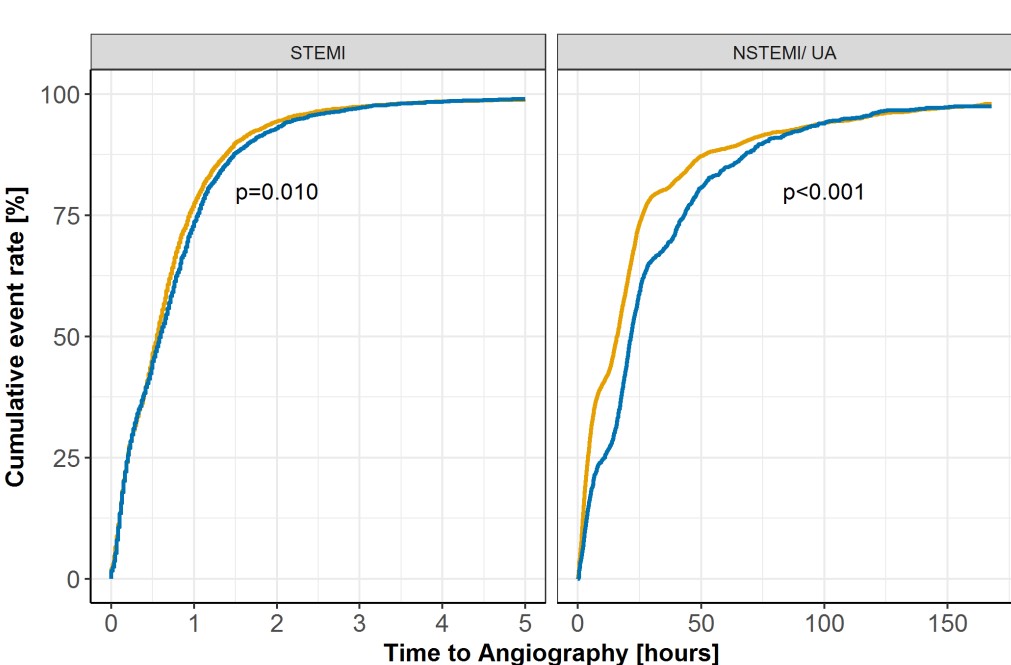

**Figure 3** Cumulative event rate for time to coronary angiography following weekend versus weekday admission across ST-elevation myocardial infarction (STEMI) and non-ST-elevation myocardial infarction/unstable angina (NSTEMI/UA) subgroups.

the two groups, at 95.5% for both weekday and weekend admission (adjusted OR 0.96, 95% CI 0.68 to 1.24).

## DISCUSSION

The findings of the current study can be summarised as follows: (1) in current UK practice, patients admitted to three tertiary cardiac centres during weekends do not present with a higher baseline risk than weekday patients,

(2) there was no difference in short-term or long-term survival between weekends and weekday groups, (3) for NSTEMI patients, there was a delay in angiography for patients admitted at a weekend and (4) other metrics of care quality such as admission under a cardiologist, performance of echocardiography and discharge medications were not adversely affected by a weekend admission.

The lack of a weekend effect in the treatment of an ST elevation myocardial infarction in a contemporary UK

**Table 5** Propensity score (PS)-adjusted HR from time-dependent Cox proportional hazards models for time to coronary angiography in patients with ST-elevation myocardial infarction (STEMI) and non-ST-elevation myocardial infarction/unstable angina (NSTEMI/UA).

| Time to coronary angiography* | **STEMI** | | |
|---|---|---|---|
| | **Mean time-to-angiography weekday, (hours)** | **Mean time-to-angiography weekend, (hours)** | **PS-adjusted HR (95% CI)** |
| 0–1 hour | 0.55 | 0.57 | **0.89 (0.84 to 0.94)** |
| 1–2 hours | 1.52 | 1.53 | 0.99 (0.87 to 1.11) |
| 2–5 hours | 3.27 | 3.11 | 1.25 (1.00 to 1.56) |
| | **NSTEMI/UA** | | |
| 0–2 hours | 1.89 | 1.94 | **0.54 (0.42 to 0.69)** |
| 2–24 hours | 15.7 | 18.3 | **0.71 (0.65 to 0.77)** |
| 24 hour–4 days | 56.2 | 53.9 | **1.22 (1.09 to 1.36)** |
| 4–7 days | 139.9 | 137.0 | 0.94 (0.68 to 1.29) |

*The time-windows for the time-dependent Cox proportional hazard model were made to satisfy the proportionality assumption throughout time. In the case of NSTEMI/UA, we selected 24 hours as one of the time cut-offs to reflect the European Society of Cardiology guidelines for high-risk patients.
Bold items indicate significant effects at the 5% level.

cohort is not surprising. The most recently published MINAP data for 2013/14 demonstrates that for the whole of the UK, 98.5% of patients with ST elevation received primary PCI, with few patients receiving thrombolysis.[22] Among its many benefits, with respect to day of the week, a primary PCI pathway is a consultant-led and consultant-delivered service, which is undertaken regardless of the time of day, or day of the week. Therefore, STEMI treatment in the UK is treated on a 24/7 basis and unless the patients admitted at a weekend are of a higher-risk profile, then it would not be expected that a weekend admission would be disadvantageous to their outcomes.

However, pathways for the treatment of NSTEMI/UA are markedly different in the UK to those for STEMI. The findings of the current study identify differences in patient treatment with respect to their timings, but not regarding patient survival. The lack of a mortality difference for a weekend NSTEMI admission may, in part, be due to the high percentage of cases admitted under the care of a cardiologist regardless of day of admission (>96% for both cohorts). Consequently, the patients may well have received optimal care in a timely fashion. Crucially, review by a cardiologist early in a patient's admission can identify patients who are unstable or deemed to be high-risk patients with NSTEMI and these can be prioritised and fast tracked to the catheter-lab along a pathway similar to primary PCI. Early identification of high-risk patients with NSTEMI by the admitting cardiologist and performing coronary angiography within 2 hours, regardless of the day of the week, can ensure that this risk is ameliorated by invasive management.[23] Indeed, several UK centres have developed direct transfer protocols to reduce time-to-angiography for patients with NSTEMI (the HAC-X group).[24] The optimal timing of angiography is less clear for intermediate-risk NSTEMI cases, with a lack of consistency in published trials.[25–27] In a recent meta-analysis of 10 time-to-angiography trials, there was no difference in mortality between patients receiving angiography early (within 24 hours) or late (24–86 hours).[28] Therefore, although the European Society of Cardiology guidelines suggest that the high-risk cohort receives angiography within 24 hours, they recommend that the intermediate-risk cohort receives angiography within 72 hours.[5] In the current study, the proportion of patients with NSTEMI/ UA receiving angiography within both 24 and 72 hours was significantly lower for weekend admissions compared with weekday admissions.

Although the current study does not demonstrate a weekend-effect on patient outcomes, it does demonstrate an effect on length of stay, which is increasingly used as an endpoint to monitor quality of service. The HAC-X group demonstrated that a direct transfer protocol for angiography in patients with NSTEMI reduced the length of stay by 6 days per admission.[24] Therefore, 7 day working (and with it angiography) may not improve patient survival after an ACS, but it is likely to significantly reduce their length of stay. A prolonged hospital admission has been closely associated with iatrogenic complications such as infection and venous thrombosis.[29 30] Hence, as well as having significant implications on bed utilisation, efficiency and costs, performing angiography on patients with NSTEMI at the weekend is likely to improve patients' experience and reduce their potential hospital-acquired morbidity. Speculatively, the longer length of stay for weekend admissions might explain the finding of higher rates of echocardiograms performed at weekends (or vice versa).

Several factors might explain the differences in the results of previous studies with the current one. First, the data analysed in the current study is derived from the period 2010 to 2016 and hence reflects contemporary ACS management. Current medical practice has evolved greatly over the last decade. Therefore, the findings of older studies will not necessarily be expected to be relevant or similar to a more contemporary cohort.[6–8] Second, the three centres involved in the current study are large university teaching hospitals with a daily review of patients by a senior cardiologist, and 24/7 access to echocardiography and cardiac catheterisation. Data that includes significant numbers of admissions to smaller hospitals without 24/7 cardiology services might lead to differing outcomes for patients admitted during the weekends compared with a weekday. Finally, differing service provision, patient populations, demographics and healthcare systems might also lead to conflicting results of previous studies with the current study.

The strengths of the current study are that it is an analysis of contemporary practice, the data are derived from a robust national database and the three centres involved are large tertiary interventional centres with high numbers of patients. However, as with any retrospective analysis, there are limitations and the robustness of the conclusions are directly related to the quality of data entered. Unmeasured confounders inherent in observational studies of this nature potentially influence any conclusions. Additionally, this study included only those patients with a discharge diagnosis of STEMI or NSTEMI/UA; consequently, outcomes might differ if the cohort was defined based solely on admission diagnosis. Finally, our findings might not generalise to other hospitals or healthcare systems due to heterogeneity in patient demographics, the healthcare system and the hospitals involved.

## CONCLUSIONS

In regional, university cardiac centres, with specialist cardiology services in the UK, patients with ACS had similar mortality and process of care when admitted at a weekend compared with a weekday. Delays to time-to-angiography occurred in patients with NSTEMI/UA patients admitted at the weekend compared with weekdays, but this did not influence survival.

**Author affiliations**
[1]Faculty of Biology, Medicine and Health, Farr Institute, University of Manchester, Manchester Academic Health Science Centre, Manchester, UK
[2]Keele Cardiovascular Research Group, Centre for Prognosis Research, Institutes of Applied Clinical Science and Primary Care and Health Sciences, Keele University, Stoke-on-Trent, UK
[3]Department of Cardiology, University Hospital of Wales, Cardiff, UK
[4]Department of Cardiology, Freeman Hospital and Institute of Cellular Medicine, Newcastle University, Newcastle-upon-Tyne, UK
[5]Department of Cardiology, Royal Stoke Hospital, University Hospital North Midlands, Stoke-on-Trent, UK

**Contributors** TK, AGZ and MAM contributed to the acquisition of the data for the analysis. GPM, TK, MS and MAM made substantial contributions to the concept of the work in addition to performing the analysis. GPM and TK drafted the initial version of the manuscript. All authors interpreted the results, revised the paper critically for important intellectual content and approved the final version of the paper. All authors agreed to be accountable for all aspects of the work.

**Funding** This work was funded by the Medical Research Council through the Health e-Research Centre, University of Manchester (MR/K006665/1) and a grant through the North Staffordshire Heart Committee.

**Competing interests** None declared.

**Ethics approval** The National Institute for Cardiovascular Outcomes Research (NICOR) which includes the MINAP database (Ref: NIGB: ECC 1-06 (d)/2011) has support under section 251 of the National Health Service Act 2006 to use patient information for medical research without informed consent. Further ethical approval was not required under current National Health Service research governance arrangements, as all data analysed in the study was pseudonymised and contained no patient identifiable information.

**Provenance and peer review** Not commissioned; externally peer reviewed.

**Data sharing statement** The statistical code used to analyse the data is available upon reasonable request by emailing MAM (corresponding author).

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
