## [Reviewer comments · BMJ Open]

ARTICLE DETAILS

TITLE (PROVISIONAL)	Effect of Weekend Admission on Process of Care and Clinical Outcomes for the Management of Acute Coronary Syndromes: a retrospective analysis of three UK centres
AUTHORS	Martin, Glen; Kinnaird, Tim; Sperrin, Matthew; Anderson, Richard; Gamal, Amr; Jabbar, Avais; Kwok, Chun; Barker, Diane; Heatlie, Grant; Zaman, Azfar; Mamas, Mamas

VERSION 1 – REVIEW

REVIEWER	Sahil Agrawal St. Luke's University Health Network, Bethlehem, PA, USA
REVIEW RETURNED	03-Apr-2017

GENERAL COMMENTS	Overall, the study investigates an important question and the manuscript is well written. The manuscript describes findings of a retrospective study to evaluate the 'weekend effect' in a contemporary ACS population I have the following questions/ concerns # Introduction line 1 Do the authors mean to say revascularization instead of intervention? # Methods Why did the authors choose data from the 3 hospitals for inclusion in this study? why were other hospitals excluded? # How were ACS patients identified? ICD codes or chart review or other processes? # Statistical analysis Authors do not define a p value for level of significance # Results It appears that weekend admission were more likely to have STEMI
--

	at presentation, and more likely to have enzyme elevations. Do the authors have a reasoning for this? Were there more UA patients in the non STEACS group admitted on the weekend? Could this be due to a longer time to medical contact in this group of patients? # I see a discrepancy in p value in results section and KM curves in figure 2 # It appears that there was some differences in time of angiography in STEMI patients b/w weekend and weekday admission. Can the authors provide information on if there were any differences in door to balloon times?
--	--

REVIEWER	Andrew Partington South Australian Department for Health & Ageing, Australia
REVIEW RETURNED	11-Apr-2017

GENERAL COMMENTS	The submitted manuscript (bmjopen-2017-016866), provides a good analysis of the impact of an admission during in/out-hours, on a suite of process and outcomes of care metrics. The paper is well written and presented, though I am unconvinced of the true comparability of the week day vs. weekend cohorts given the diagnosis definitions used. Further, some of the key results (particularly in Tables 4 and 5) could be clarified for interpretability. Particular points:  - The study aim talks of examining outcomes following “presentation with STEMI or NSTEMI/UA”. I note though that the study design utilised “discharge diagnosis” to identify the cohort, rather than an admitting/preliminary diagnosis or presenting problems suggestive of ACS. Considering the known potential for significant differences in the assessment of clinical risk and diagnostic categorisation (i.e., a false-negative preliminary and discharge diagnosis of Chest Pain (R07), rather than NSTEMI (I20)), which may both drive but also be driven by different structural (i.e., workforce or physical capacity) factors inherent within week and weekend admissions. To make a true determination on the impact of in/out-hour admissions on the flow-on care processes and outcomes, we would want to also consider the impact of differences in diagnostic accuracy and comparability of the two cohorts. This is also particularly important, when comparing the rates of diagnostic procedures, such as angiography, prior to patients receiving their discharge diagnosis and among a cohort who may be ruled-in or out of a STEMI or NSTEMI/UA. This would seem to be a significant limitation and I wonder whether the authors recognised this within their study aim, but failed to structure/capture the necessary data. - The results report ... “After 4 days, the length of stay between weekday and weekend admission was comparable” – though Table 4 indicates that the proportion of total discharges is comparable for a group discharged between 4 and 50 days. This is a different result, and we are unable to determine the ALOS within this broad category. These results should be presented differently,
---

	especially considering as the significant difference in the immediately preceding subgroup of 1 – 4 days. - The sub-groups in Table 5 seem arbitrary – I wonder whether the relative volume of instances within these 0-1 vs. 1-2 hours groups, would impact on the aggregate mean. The authors speculate within the discussion that ... “Speculatively, the longer length of stay for weekend admissions might explain the finding of higher rates of echocardiograms performed at weekends” – one might more reasonably expect that the relationship is instead, the other way around.
--	--

REVIEWER	Karice Hyun The George Institute for Global Health, Australia.
REVIEW RETURNED	01-May-2017

GENERAL COMMENTS	This is an interesting paper that explored the effect of weekend admission on in-hospital care and outcomes. I have a few minor comments:  1. Comparison of the mortality rate was not reported in the results section of the abstract but reported in the conclusion. I think it should be included in the results also. 2. Was intraclass correlation calculated? Was there clustering effect within centres to be controlled for? If there were clustering effect and were not accounted for, should this not be included in the limitation? 3. Please include the model used for outcomes such as prescription of medications. 4. Page 7, line 58, "calculated" should be reworded to "estimated". 5. The following sentence is quite confusing: "The Cox proportional hazards model included weekend indication as the covariate and was stratified by PS strata, thereby allowing the baseline hazard to vary across strata." (Page 8, lines 1-5). Does this mean PS groups were also adjusted? 6. Please spell out abbreviations at first use and use abbreviations throughout the rest of the manuscript.
---

VERSION 1 – AUTHOR RESPONSE

Reviewer 1:

Overall, the study investigates an important question and the manuscript is well written. The manuscript describes findings of a retrospective study to evaluate the 'weekend effect' in a contemporary ACS population. I have the following questions/ concerns:

1) Introduction line 1: Do the authors mean to say revascularization instead of intervention?
 Author response: Indeed, the term “revascularization” is more appropriate in this context than “intervention”. We have amended the text to reflect this.

2) Methods: Why did the authors choose data from the 3 hospitals for inclusion in this study? Why

were other hospitals excluded?

Author response: The authors thank the reviewer for this comment. Unfortunately, while MINAP is a national registry, this study only had access to three contributing centres. Consequently, we did not explicitly define any inclusion/exclusion criteria for the three hospitals. The following text has been amended in the methods (page 6, line 7-9) to clarify this point:

“This study had access to data from three contributing centres, namely the Freeman Hospital (Newcastle-upon-Tyne), the Royal Stoke Hospital (Stoke-on-Trent) and the University Hospital of Wales (Cardiff).”

3) Methods: How were ACS patients identified? ICD codes or chart review or other processes?

Author response: We thank the reviewer for raising this point. Each contributing centre is responsible for inputting their own data within MINAP (based on commonly agreed definitions and options for each variable). As such, we identified ACS patients through the information entered in the discharge diagnosis variable, which has the following possible options: “Myocardial infarction (ST elevation)”, “Threatened MI”, “Acute coronary syndrome (troponin positive)/ nSTEMI”, “Acute coronary syndrome (troponin negative)”, “Chest pain of uncertain cause”, “Myocardial infarction (unconfirmed)”, “Other diagnosis”, “Takotsubo Cardiomyopathy”, and “PCI related MI”. We have added the following text to the methods to clarify this (page 6, line 6-7):

“Each centre is responsible for data entry into MINAP, based on agreed definitions and options for each variable.”

This added text supports that already given on page 6 line 15-17:

“This retrospective analysis included all patients who had a discharge diagnosis of an ACS and were sub-classified into ST elevation myocardial infarction, non-ST elevation myocardial infarction, or unstable angina”.

4) Statistical analysis: Authors do not define a p value for level of significance.

Author response: We take the level of significance to be 5% and now indicate that 95% confidence intervals are reported for each estimate (page 8, line 13). Additionally, bold items within tables 2, 3, 4 and 5 indicate significant effects at the 5% level.

5) Results: It appears that weekend admission were more likely to have STEMI at presentation, and more likely to have enzyme elevations. Do the authors have a reasoning for this? Were there more UA patients in the non-STEACS group admitted on the weekend? Could this be due to a longer time to medical contact in this group of patients?

Author response: Indeed, based on Table 1, the weekend group included more STEMI cases, and elevated enzymes were observed more in weekend admitted patients. However, these are crude/unadjusted comparisons of baseline characteristics across weekday/weekend admissions. Higher proportions of STEMI and elevated enzymes in the weekend group could simply be down to chance from such a large dataset, rather than any underlying pathological factors.

The reviewer’s suggestion around differences in time between symptom onset to seeking medical care between weekend/ weekday admissions is plausible and may, in part, account for the observation that the weekend group were more likely to have enzyme elevations, although this is entirely speculative.

6) Results: I see a discrepancy in p value in results section and KM curves in figure 2.

Author response: After re-examining the results section, the authors cannot see the discrepancy indicated by the reviewer. The p-values given in the text are $p=0.569$ for the whole cohort (page 9, line 19), $p=0.032$ for the STEMI subgroup (page 9, line 22) and $p=0.103$ for the NSTEMI/UA subgroup (page 9, line 23). These correspond to those given in Figure 2.

7) Results: It appears that there was some differences in time of angiography in STEMI patients b/w weekend and weekday admission. Can the authors provide information on if there were any differences in door to balloon times?

Author response: We would like to thank the reviewer for highlighting this important point. We have now added information on door-to-balloon time for those STEMI patients who had an initial reperfusion treatment of primary PCI. We tested for significant differences in exceeding the ACC and ESC guidelines across weekend and weekday groups, and we added the following text into the results section (page 12, line 1-7):

“Similarly, we examined door-to-balloon time in those STEMI patients who underwent primary PCI as the initial reperfusion treatment and where time of reperfusion was also available. Here, door-to-balloon time was longer than 90-minutes (as recommended by the American College of Cardiology) in 6.95% of weekday admissions, compared with 9.12% of weekend admissions ($p=0.004$). Likewise, the European Society of Cardiology guideline of door-to-balloon time within 120 minutes was exceeded significantly more in weekend admissions (4.13%) compared with weekday admissions (2.96%) ($p=0.02$).”

Reviewer 2:

The submitted manuscript (bmjopen-2017-016866) provides a good analysis of the impact of an admission during in/out-hours, on a suite of process and outcomes of care metrics. The paper is well written and presented, though I am unconvinced of the true comparability of the weekday vs. weekend cohorts given the diagnosis definitions used. Further, some of the key results (particularly in Tables 4 and 5) could be clarified for interpretability. Particular points:

1) The study aim talks of examining outcomes following “presentation with STEMI or NSTEMI/UA”. I note though that the study design utilised “discharge diagnosis” to identify the cohort, rather than an admitting/preliminary diagnosis or presenting problems suggestive of ACS. Considering the known potential for significant differences in the assessment of clinical risk and diagnostic categorisation (i.e., a false-negative preliminary and discharge diagnosis of Chest Pain (R07), rather than NSTEMI (I20)), which may both drive but also be driven by different structural (i.e., workforce or physical capacity) factors inherent within week and weekend admissions. To make a true determination on the impact of in/out-hour admissions on the flow-on care processes and outcomes, we would want to also consider the impact of differences in diagnostic accuracy and comparability of the two cohorts. This is also particularly important, when comparing the rates of diagnostic procedures, such as angiography, prior to patients receiving their discharge diagnosis and among a cohort who may be ruled-in or out of a STEMI or NSTEMI/UA. This would seem to be a significant limitation and I wonder whether the authors recognised this within their study aim, but failed to structure/capture the necessary data.

Author response: Thank you for this comment. Our decision to use discharge diagnosis to define the study inclusion criteria was made to ensure the analysis only examined those patients truly admitted for ACS. We feel that use of discharge diagnosis to define the cohort is in line with our study aim. The final discharge diagnosis is determined by local clinicians according to ESC/ ACC guidelines accounting for presenting history, clinical examination, and the results of inpatient investigations.

After we applied the discharge diagnosis inclusion criteria, the STEMI sub-group used in the analysis was defined as those with an ECG determining treatment diagnosis of “ST segment elevation” and/or a discharge diagnosis of “Myocardial infarction (ST elevation)”; patients without such indication were allocated as the NSTEMI/UA sub-group. To clarify this, we have now added the following text to the methods section (page 6, line 17-20):

“After such inclusion criteria, the STEMI sub-group was defined as those patients with an admission and/or discharge diagnosis of ST elevation Myocardial Infarction; all other patients meeting the discharge diagnosis inclusion criteria were included in the NSTEMI/UA sub-group.”

While we agree with the reviewer that it would be interesting to explore differences in treatment pathway and subsequent outcomes across changing diagnoses, we feel this is beyond the scope of the current paper. Since, existing literature has examined the effect of changing diagnosis on mortality

in ACS patients (Wu et al. 2016; European Heart Journal: Acute Cardiovascular Care. DOI: 10.1177/2048872616661693) all estimates of weekend-effect reported in the current analysis were adjusted for admission diagnosis. To acknowledge the concerns raised by the reviewer, we have added a limitation that results might be different if one were to identify STEMI or NSTEMI/UA indication based solely on admission diagnosis (page 15, line 15-17):

“Additionally, this study included only those patients with a discharge diagnosis of STEMI or NSTEMI/UA; consequently, outcomes might differ if the cohort was defined based solely on admission diagnosis.”

2) The results report...“After 4 days, the length of stay between weekday and weekend admission was comparable” – though Table 4 indicates that the proportion of total discharges is comparable for a group discharged between 4 and 50 days. This is a different result, and we are unable to determine the ALOS within this broad category. These results should be presented differently, especially considering as the significant difference in the immediately preceding subgroup of 1 – 4 days.

Author response: Thank you for this comment. Length of stay was censored at 50 days (i.e. we did not model length of stay beyond 50 days), which is why Table 4 only reports the hazard ratios for length of stay up to this point. We appreciate that the text cited by the reviewer could have caused confusion for a reader comparing the results with Table 4. Consequently, we have now rephrased this to read the following (page 10, line 15):

“Between 4 and 50 days, the length of stay between weekday and weekend admission was comparable”

To determine the actual length of stay within the 4-50 day group (and throughout time), we have added the Kaplan-Meier plot for length of stay into the supplementary material (Supplementary Figure 1). From this, we can see that approximately 75% of patients had been discharged within 4 days. The changes in the hazard ratio (non-proportional hazards) occur within the first 4 days, which represents our decision to split into the time-windows presented in Table 4.

3) The sub-groups in Table 5 seem arbitrary – I wonder whether the relative volume of instances within these 0-1 vs. 1-2 hours groups, would impact on the aggregate mean.

Author response: On reflection, the authors agree with the reviewer that the time-windows in Table 5 do seem arbitrary. We selected the time-windows to (a) satisfy the proportionality assumption of the Cox proportional hazards models (data driven), and (b) in the case of NSTEMI to reflect the 24-hour ESC guidelines for high-risk patients. We have now added the following footnote to Table 5 that describes this:

“*: The time-windows for the time-dependent Cox proportional hazard model were made to satisfy the proportionality assumption throughout time. In the case of NSTEMI/ UA, we selected 24 hours as one of the time cut-offs to reflect the ESC guidelines for high-risk patients.”

4) The authors speculate within the discussion that...“Speculatively, the longer length of stay for weekend admissions might explain the finding of higher rates of echocardiograms performed at weekends” – one might more reasonably expect that the relationship is instead, the other way around.

Author response: We have now indicated that the interpretation could be made in either direction (page 14, line 22):

“Speculatively, the longer length of stay for weekend admissions might explain the finding of higher rates of echocardiograms performed at weekends (or vice versa).”

Reviewer 3:

This is an interesting paper that explored the effect of weekend admission on in-hospital care and outcomes. I have a few minor comments:

1) Comparison of the mortality rate was not reported in the results section of the abstract but reported in the conclusion. I think it should be included in the results also.

Author response: We would like to thank the reviewer for highlighting this. The results section of the abstract has been edited to indicate that mortality rates were similar across weekday and weekend admissions. Additionally, we have made several structural changes to the abstract to match the BMJ Open formatting criteria.

2) Was intraclass correlation calculated? Was there clustering effect within centres to be controlled for? If there were clustering effect and were not accounted for, should this not be included in the limitation?

Author response: We are unsure what intraclass correlation the reviewer suggests should be calculated. Data entry into MINAP is based on pre-specified variable definitions and should, therefore, be standardised/consistent across centres. Nevertheless, we agree that there is potential for differences in outcomes across the contributing centres. As such, the analysis did include centre indication when calculating the propensity scores, thereby adjusting the weekend admission odds/hazard ratios for a centre effect. The below text indicates this point (page 7, line 20-21): "Variables included in the logistic regression model to calculate the PS included all those given in Table 1 and an admitting centre indicator."

3) Please include the model used for outcomes such as prescription of medications.

Author response: Binary endpoints (i.e. hospital mortality, 30-day mortality, admission to cardiology ward, coronary angiography, echocardiography, prescription of medications) were modelled using logistic regression stratified by PS. Time-to-event outcomes (i.e. long-term survival, time to angio, length of stay) were modelled using Cox proportional hazards. We have clarified which endpoints were modelled through each technique within the statistical analysis section of the methods (page 8, lines 1-7):

"Within each PS strata, binary outcomes (mortality, admission to cardiology ward, coronary angiography, echocardiography, prescription of medications) were compared directly between admission day groups (using logistic regression), with the odds ratios (ORs) pooled by the Mantel-Haenszel method. For time-to-event outcomes (length of stay, time-to-angiography and long-term survival), the unadjusted survival curves were obtained using the Kaplan-Meier estimate and adjusted hazards ratios (HRs) were calculated estimated using a Cox proportional hazards model."

4) Page 7, line 58, "calculated" should be reworded to "estimated".

Author response: We thank the reviewer for noticing this, and have now changed the wording accordingly (page 8, line 7).

5) The following sentence is quite confusing: "The Cox proportional hazards model included weekend indication as the covariate and was stratified by PS strata, thereby allowing the baseline hazard to vary across strata." (Page 8, lines 1-5). Does this mean PS groups were also adjusted?

Author response: The authors acknowledge that the original wording of this sentence could lead to confusion. We meant that the PS strata were used to stratify the baseline hazard function of the Cox proportional hazards model (i.e. the Cox model was fitted within strata of the PS).

To help with understanding, we have now removed the latter part of the original sentence (page 8, line 8-9):

"The Cox proportional hazards model included weekend indication as the covariate and was stratified by PS strata."

6) Please spell out abbreviations at first use and use abbreviations throughout the rest of the manuscript.

Author response: All abbreviations in the manuscript have now been either removed if they were only used once, or spelt out at first use.

VERSION 2 – REVIEW

REVIEWER	Sahil Agrawal St. Luke's University Health Network
REVIEW RETURNED	30-Jun-2017

GENERAL COMMENTS	I would like to thank the authors for the revisions. Again, the manuscript is well written, and attempts to answer an important question, albeit with some limitations. I would like the authors to consider reframing the results section where they discuss the length of stay. It is somewhat confusion. Would it not be better to just median lengths of stay instead of stratifying them. If more patients in one group are discharged early (<1d), by reverse would be true by default.
--

REVIEWER	Karice Hyun The George Institute for Global Health, Australia
REVIEW RETURNED	24-Jun-2017

GENERAL COMMENTS	Thank you for making the edits. I have no further comments.
---

VERSION 2 – AUTHOR RESPONSE

Reviewer 1:

I would like to thank the authors for the revisions. Again, the manuscript is well written, and attempts to answer an important question, albeit with some limitations.

1. I would like the authors to consider reframing the results section where they discuss the length of stay. It is somewhat confusion. Would it not be better to just median lengths of stay instead of stratifying them. If more patients in one group are discharged early (<1d), by reverse would be true by default.

Author response: The authors thank the reviewer for this comment. In response, we have rephrased the length of stay results section, to try to make this easier to follow. Specifically, the text now reads (pg. 10):

“Weekend admission was associated with significantly lower hazards of being discharged within one-day (HR 0.72, 95% CI: 0.66, 0.78), but significantly higher hazards of being discharged between 1 and 4 days (HR 1.13, 95% CI: 1.08, 1.18) (Table 4).”

Our decision to stratify length of stay was made to satisfy the proportionality assumption of the Cox proportional hazards model, thereby allowing the hazard ratio of being discharged to vary through time. For example, the results indicate that weekend admissions were significantly less likely to be discharged within one-day, but those patients who were not discharged by one-day were subsequently more likely to be discharged between 1 and 4 days. We feel that analysing length of stay in terms of median values would fail to capture this dynamic process. Additionally, the median length of stay for each group can be obtained from the corresponding Kaplan-Meier plot (Supplementary Figure 1).

Reviewer 3:

Thank you for making the edits. I have no further comments.
Author response: Many thanks for your review.

VERSION 3 – REVIEW

REVIEWER	Sahil Agrawal St. Luke's University Health Network, USA
REVIEW RETURNED	11-Aug-2017

GENERAL COMMENTS	No further comments. Thank you for incorporating my suggestions. Good luck!
--